# Prevalence and Determinants of COVID-19 Vaccine Acceptance in South East Asia: A Systematic Review and Meta-Analysis of 1,166,275 Respondents

**DOI:** 10.3390/tropicalmed7110361

**Published:** 2022-11-09

**Authors:** Theo Audi Yanto, Nata Pratama Hardjo Lugito, Lie Rebecca Yen Hwei, Cindy Virliani, Gilbert Sterling Octavius

**Affiliations:** Department of Internal Medicine, Universitas Pelita Harapan, Tangerang 15811, Indonesia

**Keywords:** South East Asia, COVID-19 vaccine acceptance, prevalence, determinants, Indonesia, Singapore, Thailand, Myanmar, Vietnam, Malaysia

## Abstract

Despite its importance in guiding public health decisions, studies on COVID-19 vaccination acceptance and its determinants in South East Asia (SEA) are lacking. Therefore, this study aims to determine the prevalence of COVID-19 vaccine acceptance and the variables influencing the vaccine’s acceptance. This review is registered under PROSPERO CRD42022352198. We included studies that reported vaccination acceptance from all SEA countries, utilising five academic databases (Pubmed, MEDLINE, Cochrane Library, Science Direct, and Google Scholar), three Indonesian databases (the Indonesian Scientific Journal Database, Neliti, and Indonesia One Search), two pre-print databases (MedRxiv and BioRxiv), and two Thailand databases (ThaiJo and Thai-Journal Citation Index). The analysis was conducted using STATA 17.0 with metaprop commands. The prevalence for COVID-19 vaccination acceptance in SEA was 71% (95%CI 69–74; I2 99.87%, PI: 68.6–73.5). Myanmar achieved the highest COVID-19 vaccination acceptance prevalence, with 86% (95%CI 84–89), followed by Vietnam with 82% (95% CI 79–85; I2 99.04%) and Malaysia with 78% (95%CI 72–84; I2 99.88%). None of the ten determinants studied (age, sex, education, previous COVID-19 infections, smoking and marriage status, health insurance, living together, chronic diseases, and healthcare workers) were significantly associated with acceptance. This result will be useful in guiding vaccination uptake in SEA.

## 1. Introduction

Severe acute respiratory syndrome coronavirus 2 (SARS-CoV-2), a strain that caused coronavirus disease in 2019 (COVID-19), was discovered almost three years ago, and it is still causing a pandemic worldwide [1]. The pandemic has had enormous impacts on the world from medical sectors, such as an increased number of deaths [2,3,4,5,6] and mental health issues [7], to the economy [8,9], as well as its adverse effects on education [10].

Numerous medical treatments have been proposed to combat COVID-19 [11,12], yet non-medical interventions, such as prohibitions of large-scale gatherings and mask mandates, are still the best way to prevent and curb the spread of COVID-19 [13,14]. In terms of public health, the rapid COVID-19 vaccine development and campaigns are significant medical successes, and were touted as game changers in this pandemic [15]. Through rigorous randomised clinical trials, COVID-19 vaccines proved to be safe and effective among children and adults [16,17,18].

A vaccine can only be successful if it is administered adequately in a population. However, due to misinformation and misunderstandings, COVID-19 vaccination hesitancy or outright refusal rose and impeded COVID-19 vaccination [19]. Vaccine acceptance is the opposite of vaccine refusal, and it is one of the significant determinants influencing vaccine uptake [20,21]. It has been hypothesised that vaccine uptake is determined by access and acceptance, further stressing the importance of vaccine acceptance [22]. Thus, unearthing the reasons behind acceptance and knowing the current COVID-19 vaccination acceptance rate will help the government and medical workers strategise in order to increase the COVID-19 vaccination uptake.

Vaccine acceptance, hesitancy, and refusal are described as a state of a continuum with complex interplay and variables involved that influence them. Contextual determinants (historical, political, and sociocultural influences as well as communication and media environment), individual determinants (sociodemographic characteristics, knowledge and attitude, past experiences with health and vaccination services, and trust in the health system and healthcare providers), organisational determinants (availability and quality of vaccination services, health staff motivation and attitudes, as well as vaccine-specific issues) are some of the factors that affect vaccine hesitancy and refusal [23]. The reason why addressing vaccine refusal and hesitancy is important is that targeting specific interventions will not convince those who are hesitant or refusing vaccines to accept vaccines instantly. This problem is also compounded by limited evidence on how to best address vaccine hesitancy. Currently available techniques must be carefully adjusted for the target audience, the reasons for reluctance, and the particular situation in a given population [24]. 

However, numerous pieces of literature do not seem to agree on what determinants influence vaccine acceptance. Numerous pieces of literature mention age, previous history of COVID-19 infection, hospitalisation due to COVID-19, belief in the safety and efficacy of COVID-19 vaccination, jobs, body mass index, knowledge, education status, marital status, possessing health insurance, sex, having chronic conditions, and living together as determinants that affect COVID-19 vaccine acceptance. Nonetheless, the results are conflicting, with one study mentioning the positive impact of one determinant while other studies will find an opposite or null effect of that same determinant [25,26,27,28,29,30].

Despite numerous meta-analyses studying the prevalence of COVID-19 vaccination acceptance, we could not identify a single one focusing on South East Asia (SEA) [27,31,32,33]. Even though SEA countries contribute to 8.58% of the world population [34], studies focusing on COVID-19 vaccination acceptance and its determinants in SEA are surprisingly rare, if they exist at all. One meta-analysis that we could find is by Norhayati et al., who conducted a worldwide meta-analysis and concluded that vaccine acceptance was high in SEA, at 61%. However, only eight studies were included in the SEA analysis. Four studies originated from India or Bangladesh, and these two countries are not part of SEA. The other four studies originated from Indonesia and Malaysia, which are not adequate representations for all SEA countries [35].

With the non-existent studies focusing on SEA regarding COVID-19 vaccination acceptance and its determinants, despite its importance in guiding public health decisions, we decided to conduct a systematic review and meta-analysis. This study aims to determine the prevalence of COVID-19 vaccine acceptance, with a secondary aim of determining the variables that influence its acceptance.

## 2. Materials and Methods

### 2.1. Eligibility Criteria

The authors adhered to The Preferred Reporting Items for Systematic Review (PRISMA) 2020 guidance [36]. The protocol of this review was registered on the International Prospective Register of Systematic Reviews (PROSPERO) database with the registration number CRD42022352198.

The studied population consisted of everyone eligible for COVID-19 vaccination, including children and pregnant women. We included studies that reported vaccination acceptance. The primary outcome of this study is to calculate the point prevalence of COVID-19 vaccine acceptance in SEA countries. A sub-group analysis would be conducted according to types of paper (peer-reviewed journals vs. grey literature), the timing of vaccine rollout relative to when the study was conducted, questionnaire type, whether the study was referring to COVID-19 vaccine boosters or not, sampling methodology, data collection method, quality of the study, and risk of bias. If studies were conducted in two different years (e.g., 2020–2021), the year with more calendar days in which the study was conducted would be chosen—Appendix A lists the timing of vaccine rollout. As for the questionnaire type, there are two major classifications, including those using the Likert scale and the other one using “yes or no questions.” The way the questions were phrased varied between studies and contexts. However, most questions were phrased as “If COVID-19 vaccines are available now, will you accept them?” The secondary outcome of this study is to analyse the determinants of COVID-19 vaccine acceptance in SEA. Some of the factors analysed included age, sex, education status, smoking status, previous COVID-19 infection, possessing health insurance, living alone or together, being a healthcare worker (HCW), marital status, and presence of chronic diseases. There were no interventions or control groups in this meta-analysis.

The readiness to receive vaccinations, acceptability of a vaccine, desirability, demand, and favourable sentiments regarding the administered vaccines are all definitions of vaccine acceptance used in this meta-analysis [35]. The inclusion criteria were cross-sectional studies or observational studies published in any language that were published from 2020 onwards. We extracted all valid results for repeated cross-sectional studies and treated them as two separate datasets from one study. Hence, there would be more datasets compared to the number of studies. Studies that used mixed-methodology (qualitative and quantitative) were also included. In order to ensure that all data were fully extracted, all Appendix A were searched. We also included grey literature such as conference abstracts, theses, conference papers, government or other independent research bodies, and dissertations. The exclusion criteria of this study were case reports, case series, cohort studies, reviews, animal studies, and studies that were conducted outside of SEA. We also excluded studies with <50 samples in order to maintain the stability of the prevalence pool [37]. Another exclusion criterion would be studies that reported vaccination rate and not vaccination acceptance. Lastly, studies that extracted secondary data from social media, such as comments or tweets, would also be excluded. There were no restrictions on language, and non-English and non-Indonesian studies would be translated using Google Translate. We also hired professional translators proficient in the specific language needed to interpret the article correctly. Corresponding authors were also contacted to help us correctly extract the necessary data from their articles. In order to ensure saturation of the literature, citations from review studies were scoured. We also performed citation and hand-searching from excluded articles in order to ensure that all available studies were included.

### 2.2. Search Strategy and Study Selection

The literature search started on 25 August 2022 and ended on the same day. We searched five academic databases: Pubmed, MEDLINE, Cochrane Library, Science Direct, and Google Scholar. Three Indonesian-specific databases were also utilised in order to increase literature saturation: the Indonesian Scientific Journal Database (ISJD), Neliti, and Indonesia One Search. Medrxiv and Biorxiv were scoured for grey literature that was not peer-reviewed yet. Studies from these two databases were searched for the peer-reviewed version before confirming the grey literature status. Two Thailand-specific academic databases, ThaiJo and Thai-Journal Citation Index, were also utilised. The search was conducted using English and Bahasa Indonesia. The keywords used were related to COVID-19 (“COVID-19”, “SARS-CoV-2”), vaccine stances (“vaccine acceptance”, “vaccine hesitancy”, “vaccine refusal”, “vaccine uptake”), and countries (“South East Asia”, “Brunei Darussalam”, “Indonesia”, “Singapore”, “Malaysia”, “Timor Leste”, “Philippine”, “Myanmar”, “Cambodia”, “Thailand”, “Vietnam”, “Laos”). The Medical Subject Heading (MeSH) terms for each database can be seen in Appendix A. All records were entered into the Rayyan software, which automatically recognised duplicates and manually screened them [38]. This software also allowed authors to collaborate in selecting relevant studies. Two independent authors conducted the initial search (GSO and CV), importing all the findings into Rayyan software. Another author (LRYH) cross-checked the initial searches. These three authors independently screened all available studies. Conflicts were resolved by discussion with the experts (TAY and NPHL). In the case of studies with overlapping time points from the same dataset, we chose the data that provided us with the most available information.

### 2.3. Data Extraction and Quality Assessment

Data extraction was carried out independently by two authors separately (GSO and LRYH), and then reviewed by the third author (CV) in order to ensure accuracy. We extracted relevant information such as study identification (author and year of publication), study characteristics (location, study design, and study period), and vaccine acceptance rate (total population studied and the number of respondents who were vaccine accepting). When studies used different theoretical vaccine efficacy to determine vaccine acceptance, we chose those with the highest suggested vaccine efficacy in each study. However, one study had a 100% vaccine acceptance rate with 80% vaccine efficacy; therefore, we opted for the second-highest theoretical vaccine efficacy level [39].

The Newcastle–Ottawa scale (NOS) for cross-sectional studies was implemented in order to assess the quality of the studies. A score of 7–9 on NOS implied the study indicated a good quality, 4–6 indicated a moderate or fair quality, and a score of 0–3 meant that the study had poor quality [40]. If one study had multiple datasets, those datasets would be judged based on the quality of that original study. As for the risk of bias, we used Joanna Briggs’ Institute (JBI) criteria for prevalence studies. Studies with a score of 0–3 were considered low risk, a score of 4–6 was considered moderate risk, and a score of 7–9 was considered high risk [41]. Three reviewers (GSO, CV, LRYH) independently assessed the NOS and JBI scale, and any discrepancies were sorted with the experts (TAY and NPHL) until a consensus was attained. If missing or further data were needed, corresponding authors were sent an inquiry email or a message via ResearchGate. Studies that were too short to be assessed, such as abstracts or posters, would not be graded for NOS and JBI.

### 2.4. Data Synthesis

We computed the point prevalence by dividing the number of respondents who identified themselves as vaccine accepting by the total number of subjects in that study [42]. The analysis was conducted using STATA software (Version 17.0, StataCorp, College Station, TX, USA), and Metaprop was the command of choice for prevalence calculation. DerSimonian and Laird’s random-effect model was chosen, and we calculated the 95% confidence interval (CI) using the Clopper–Pearson method [43]. We used prediction intervals to assess heterogeneity [44], and between-study heterogeneity was explored with a Galbraith plot [45]. Small-study effects would be assessed with funnel plot analysis if there were more than ten studies included [46], namely Begg and Mazumdar’s test for rank correlation [47] and Egger’s test for a regression intercept [48]. Trim-and-fill analysis would be conducted if there were an asymmetry in the funnel plot [49]. Random effects with Hartung–Knapp–Sidik–Jonkman were used to analyse the determinants of COVID-19 vaccination acceptance with prediction intervals, using the metaprop command [50,51]. If cells with 0 values existed, we used continuity correction by adding 0.5 to that blank cells [52].

## 3. Results

We identified 173,361 manuscripts, and 1826 of these articles were duplicates. A total of 166,749 records were eliminated after the title and abstract assessment, with 4786 articles sought for a full assessment. A total of 71 articles were included in the analysis. We also found 40 articles from citation-searching and hand-searching, which resulted in an additional 38 datasets. In total, 109 studies were available for systematic review and meta-analysis (Figure 1). The NOS and JBI scores are presented in Appendix A, respectively. Notable exclusions and the reasons for exclusions are included in Appendix A.

There were 1,166,275 respondents included in this review, and 798,850 respondents identified themselves as COVID-19 vaccine acceptors. There were a total of 135 datasets, with Malaysia contributing the most datasets (n = 26), followed by Indonesia, with 25, and Thailand, with 22. There was only one study in Brunei Darussalam, while no studies were found in Cambodia. There were 111 datasets published in peer-review journals, while 24 were published in grey literature, such as websites, government documents, or posters (Appendix A).

Most datasets published came from studies which were conducted before the vaccine rollout (n = 71) and studied the first two vaccine shots (n = 126). The majority used a non-probability (n = 106) sampling methodology, and 118 datasets were obtained with questionnaires that were not administered directly by researchers. Many datasets had good NOS (n = 67) and JBI (n = 70) scores.

The prevalence of COVID-19 vaccination acceptance was 71% (95%CI 69–74; I^2^ 99.87%, PI: 68.6–73.5). Myanmar achieved the highest COVID-19 vaccination acceptance prevalence, with 86% (95%CI 84–89), followed by Vietnam with 82% (95% CI 79–85; I^2^ 99.04%) and Malaysia with 78% (95%CI 72–84; I^2^ 99.88%). Brunei Darussalam only achieved 59% (95%CI 59–59), the lowest prevalence of COVID-19 vaccine acceptance amongst other SEA countries (Figure 2). The Galbraith plot indicates some heterogeneity, as there are a few outlier studies, and the funnel plot is asymmetric (Appendix A). Trim-and-fill analysis suggests no significant difference after adjustments. Begg and Mazumdar’s test for rank correlation gives a *p*-value of 0.0003, indicating possible evidence of publication bias. In contrast, Egger’s test for a regression intercept gives a *p*-value of <0.0001, indicating possible evidence of publication bias.

When analysed annually, there was a downward trend in COVID-19 vaccine acceptance. The rate decreased from 74% (95%CI 71–78) in 2020 to 71% (95% CI 68–74) in 2021, and plunged to 56% (95%CI 45–68) in 2022. Singapore is the only country with an upward trend of COVID-19 vaccine acceptance (65% [95%CI 42–88] in 2020 to 69% [95%CI 59–79] in 2021). Malaysia showed an initial 4% upward trend of COVID-19 vaccine acceptance between 2020–2021, before the acceptance rate fell to just 43% (95%CI 40–46) in 2022 (Figure 3). Studies conducted in Brunei Darussalam, Laos, and East Timor were conducted in the same year; hence, their trend could not be analysed.

Subgroup analysis of prevalence was conducted amongst types of papers, with a higher prevalence amongst studies published in peer-reviewed journals (73%; 95%CI 70–75) compared to grey literature, with 65% (95%CI 60–70). When the type of vaccination (booster or not booster) was assessed, the prevalence was higher amongst non-booster vaccines, with 72% acceptance (95%CI 70–74). Vaccine acceptance was also higher amongst studies conducted before vaccine rollout (72%, 95%CI 69–75), using mixed-methods sampling (77%; 95%CI 77–77), questionnaires not administered by researchers (73%, 95%CI 70–75), as well as in studies with moderate NOS criteria (74%, 95%CI 68–79) and moderate JBI risk (76%, 95%CI 71–80). Studies using the Likert scale yielded the same vaccination acceptance rate as those using yes or no questions (72%, 95%CI 68–76 for the Likert scale, and 95%CI 68–75 for yes or no) (Appendix A).

Having chronic diseases (odds ratio [OR] 0.83; 95%CI 0.67–1.03, PI 0.66–1.05) (Appendix A), being in the age range of 18–64 years old (OR 1.07; 95%CI 0.64–1.78, PI 0.63–1.08) (Appendix A), having completed at least high school (OR 1.72; 95%CI 1.01–2.92, PI 0.97–3.04) (Appendix A), being a healthcare worker (OR 1.69; 95%CI 0.96–2.96, PI 0.94–3.05) (Appendix A), having health insurance (OR 0.91; 95%CI 0.56–1.48, PI 0.55–1.51) (Appendix A), living together (OR 1.25; 95%CI 0.3–5.19, PI 0.28–5.58) (Appendix A), being married (OR 0.79; 95%CI 0.58–1.07, PI 0.57–1.10) (Appendix A), contracting the COVID-19 infection in the past (OR 1.06; 95%CI 0.7–1.62, PI 0.68–1.64) (Appendix A), being male (OR 1.13; 95%CI 0.98–1.31, PI 0.96–1.34) (Appendix A), and being a smoker (OR 1.47; 95%CI 0.96–2.26, PI 0.95–2.28) (Appendix A) were all not significantly associated with COVID-19 vaccine acceptance.

## 4. Discussion

### 4.1. Comparison of COVID-19 Vaccination Acceptance Rate with Other Studies

The COVID-19 vaccination rate in SEA countries is 71%. This number is lower compared to one meta-analysis, which discovered a vaccination acceptance rate of 70% in Europe (30 studies) and 74.6% in Asia (10 studies) [53]. Another meta-analysis found a similar rate of 70.8% (95%CI 58.12–82.25) in SEA countries, with only five studies. However, this meta-analysis included three studies from Bangladesh and one study that we excluded due to the nature of secondary data analysis from social media platforms [54]. These findings indicate that our findings on vaccine acceptance rate in SEA are comparable to other meta-analyses.

Sallam concluded in his meta-analysis that Malaysia (94.3%) and Indonesia (93.3%) had the highest COVID-19 vaccination acceptance in the world [55]. Their data contrast with ours, with only 78% vaccination acceptance for Malaysia and 67% for Indonesia. However, it should be noted that only one study was included in their meta-analysis, falsely inflating the vaccination acceptance rate [56]. Another meta-analysis by Shakeel et al. also arrived at the same result as Sallam, citing the same two studies [31].

Vietnam achieves the second-highest COVID-19 vaccination acceptance prevalence, at 82%. It is well known that Vietnam is a model success story regarding COVID-19 vaccination uptake [57]. The World Health Organization attributed this success to securing a sustainable COVID-19 vaccine supply, ensuring health system readiness and efficiency during vaccination rollout, taking into account operational considerations, providing training on safe vaccination and vaccine safety surveillance, communicating with the public to increase knowledge and influence vaccine uptake behaviour, and enhancing the information system for data collection and reporting [58,59]. It should be noted that Vietnam’s COVID-19 vaccine rollout was second to last amongst other SEA countries, with only East Timor being the last to vaccinate their own people [60,61]. Despite the late rollout and slow initial COVID-19 uptake, Vietnam achieved high COVID-19 vaccination acceptance and vaccination uptake [59].

The Philippines achieves the second lowest COVID-19 vaccination acceptance, at 61%. Much of the vaccine hesitancy in the Philippines stems from the Dengvaxia^®^ controversy. This event shows that once the public hesitates about vaccines, it becomes a mountainous challenge to win back the public’s trust and confidence in the vaccine [62,63]. Myanmar and Brunei Darussalam are the highest and lowest countries for COVID-19 vaccine acceptance, respectively. However, it is essential to note that there are only two datasets studying Myanmar and only one study for Brunei Darussalam, which may artificially increase or decrease the rate. Laos is also another country in this meta-analysis with only two datasets.

### 4.2. Trends in COVID-19 Vaccination Acceptance Rate in SEA

Overall, there was a downward trend in the COVID-19 vaccination acceptance rate from 2020 to 2022. One possible explanation for this is because of COVID-19 news fatigue. People are tired of listening to the same COVID-19 news; hence, they tend to tune out of COVID-19 news, including news regarding COVID-19 vaccines [64,65]. Another possible explanation is the easing of COVID-19 restrictions. The bans imposed during COVID-19, such as large social gatherings and traveling, have been lifted, falsely signifying to the public that COVID-19 is almost over [66]. Restriction ease will cause some people to believe that the previously mandatory COVID-19 vaccination seems unnecessary. Lastly, introducing booster shots around mid-2021 may also cause declining COVID-19 vaccination acceptance. Hesitation about the booster shots, systemic mistrust, religious issues, lack of information, and health concerns are among the cited reasons for refusing COVID-19 booster shots [67,68].

### 4.3. Findings of Sub-Group Analysis

The COVID-19 vaccine acceptance rate was higher in peer-reviewed journals than in grey literature. This finding shows that studies with a higher acceptance rate will be more likely to be published in peer-reviewed journals. Studies with a lower acceptance rate will be submitted to grey literature and may not be accepted in peer-reviewed journals [69]. Studies concerning the first two jabs reported higher COVID-19 vaccination acceptance than those focusing on booster shots. The reasoning has been stated above, but it is noteworthy that only 6 datasets focused on booster vaccines, compared to 126 datasets studying non-booster vaccines.

Sampling methodology affects the results of a study, as studies utilising probability methods achieved a lower vaccination acceptance rate compared to non-probability samples (66% vs. 74%). Therefore, this finding shows that studies with non-probability sampling may achieve results that are higher than they are supposed to be [70]. Studies conducted through web surveys are increasing due to the COVID-19 lockdown. Although dissemination is quick and low-cost, there are some concerns regarding participation bias and duplicate answers [71]. Therefore, the higher acceptance rate in studies not administered directly by researchers (i.e., web surveys) may be biased. Lastly, studies with moderate NOS and JBI achieved a higher acceptance rate than studies that scored “good” on NOS or “low risk” with JBI. Studies have shown that studies with less rigorous qualities may distort the conclusion of a study [72,73]. In our case, studies with good NOS scores and low risk according to the JBI scale have acceptance rates closer to the pooled acceptance rate than those with moderate NOS scores and JBI risk.

None of the ten variables studied are significantly associated with vaccination acceptance, due to the CIs or PIs crossing the value of one. One meta-analysis also found that sex did not affect vaccine acceptance significantly [33]. We only managed to find one meta-analysis that agreed with our findings [33], while other meta-analyses found significant associations between the variables studied and COVID-19 vaccine acceptance. This finding reiterates the importance of bias and confounding in observational studies, which may find significant associations on their own, but not in large-scale, experimental studies or meta-analyses [74]. However, we also acknowledge the fact that only randomised controlled trials (RCT) are able to confirm the nature of a finding, as observational studies tend to have conflicting results with RCTs. Due to the nature of studies conducted for this type of study, RCTs are extremely difficult to conduct, and hence, meta-analysis is one alternative to support this finding conclusively. It should be kept in mind by our readers that some studies included one of the ten variables studied here, but due to different categorisations from our meta-analysis, those studies were excluded. This nature of dichotomisation may impact the findings. We explain the effects of dichotomisation more in the following section below.

### 4.4. Limitation

There are several limitations to our systematic review and meta-analysis. This study is a meta-analysis of cross-sectional studies; hence, the evidence might not be as sound as a meta-analysis of randomised controlled trials (RCT). However, given how the studies were conducted, it is not feasible to conduct RCTs; thus, this type of meta-analysis may be the best evidence available. Another limitation lies in missing studies from Cambodia and the low number of studies from Myanmar, Laos, Brunei Darussalam, and East Timor. One contributing factor may be language issues. We cannot exclude the possibility that studies originating from those countries are published in their native languages, leading to unidentifiable studies.

Moreover, COVID-19 vaccination acceptance does not equate to vaccination uptake. While our results are roughly equal to the current COVID-19 vaccination status, such as in Indonesia (67% vaccination acceptance [VA] vs. 62.5% completely vaccinated [CV]), Malaysia (78% VA vs. 81.9% CV), and Vietnam (82% VA vs. 85% CV). Some countries have large disparities, such as in Thailand (67% VA vs. 75% CV) and Singapore (68% VA vs. 92% CV) [75]. Other factors affect vaccine uptake aside from acceptance, such as access, affordability, awareness, and activation, which we did not look into [20]. Other variables that are associated with COVID-19 vaccine acceptance, such as psychosocial factors [76], trust in government, HCWs, influential people [77], knowledge, attitude, behaviours [78], vaccine efficacy [35], income [27], and misinformation regarding COVID-19 vaccination [31], are not included in this meta-analysis, and thus no conclusions can be derived. Due to the dichotomisation nature of vaccine acceptance studies, some studies that look into the variables which we assessed might not be included because of the differences in how the categories were split (e.g., we categorised education as those who finished at least junior high school and those who do not complete junior high school). The exclusion of these studies may affect the pooled results [79].

Lastly, a sub-group analysis is a valuable tool for assessing the difference between the two groups, but it is not a confirmatory method that eliminates bias. Even though we have tried our best to assess the outcome according to variables that may affect the prevalence, we advise our readers to interpret these findings cautiously.

### 4.5. Strengths

There are, however, some strengths of our study. This meta-analysis is the first to study the prevalence of COVID-19 vaccine acceptance in South East Asia exclusively, yielding 125 datasets and 1,166,275 respondents. Aside from its large sample size, we also identified studies published in grey literature and included them in our meta-analysis. Although there are controversies surrounding grey literature, such as the fact that the quality of such publications tends to be lower compared to peer-reviewed journals, grey literature is an essential resource in systematic reviews [80]. Therefore, previously uncited papers in other meta-analyses are incorporated into our meta-analysis. Some of the grey literature is also of good quality. For example, OCTA is an independent science advice provider in the Philippines that published its COVID-19-related findings on various websites. They are a legitimate science body that disseminates valid scientific results [81]. Cochrane group also strongly advises researchers to search for grey literature thoroughly in order to minimise the impact of publication bias. Hence, the small-study effects found in our meta-analysis are unlikely to be due to publication bias alone, as we have thoroughly searched and included all available studies that we could find. Lastly, although the heterogeneity seems high by the I^2^ index, the PI is narrow and within the confidence interval. The PI tells us what the results will be when future studies are conducted, which is more clinically useful. The I^2^ index does not generally represent heterogeneity per se, and PI represents heterogeneity better than the I^2^ index [82].

## 5. Conclusions

The COVID-19 vaccination acceptance rate in SEA countries may not be adequate to achieve herd immunity. Although vaccine acceptance does not equal vaccine uptake, some of these rates translate to the current vaccination rate in some countries. Therefore, unless external forces such as government intervention or increasing access to COVID-19 vaccination are incorporated, the vaccination rate trajectory will slow down eventually [20].

There is no single variable that is significantly associated with COVID-19 vaccination acceptance. In line with this finding, we agree that COVID-19 vaccination acceptance is influenced by many complex interplays of factors specific to a particular country or region due to differences in religion, culture, or beliefs [21]. Researchers from different SEA countries should form a large research group that assesses vaccine acceptance using the same methodology while analysing them according to each country’s different cultures, backgrounds, and situations. Therefore, instead of focusing on specific sociodemographic characteristics, the next step will be to identify tailored methods to increase COVID-19 vaccination acceptance in a specific population, translating into vaccination uptake.

## Figures and Tables

**Figure 1 tropicalmed-07-00361-f001:**
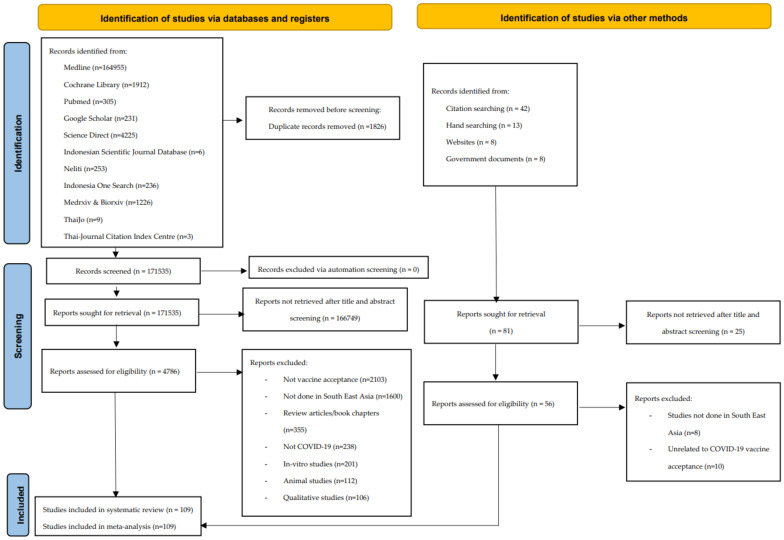
PRISMA flowchart for the selection of included studies.

**Figure 2 tropicalmed-07-00361-f002:**
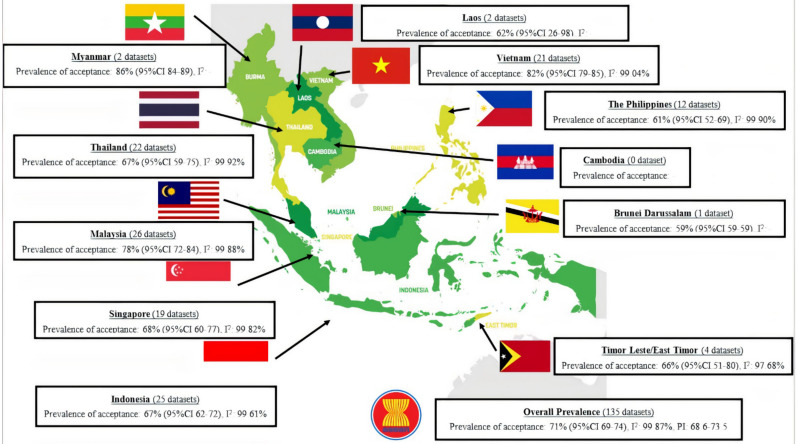
Prevalence of COVID–19 Vaccine Acceptance in South East Asia.

**Figure 3 tropicalmed-07-00361-f003:**
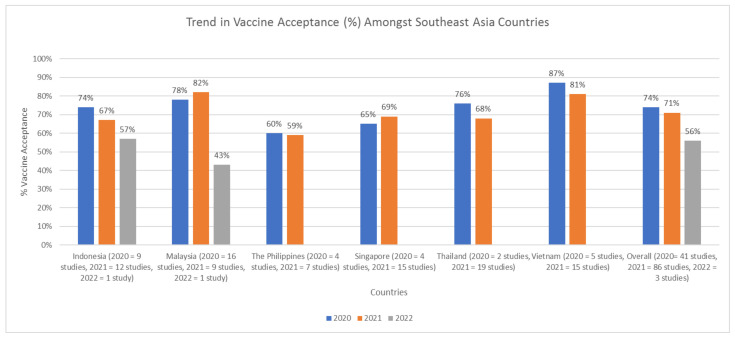
Trends in vaccine acceptance changes amongst South East Asian countries between 2020 and 2022.

## Data Availability

Data generated in this study is available by contacting the first author, T.A.Y., if requested reasonably.

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
