# Peer review of "Prevalence and Determinants of COVID-19 Vaccine Acceptance in South East Asia: A Systematic Review and Meta-Analysis of 1,166,275 Respondents"

_tropicalmed, 2022, doi:10.3390/tropicalmed7110361_

Round 1
Reviewer 1 Report
Dear authors,
The topic of this review is very significant given the gap in the current literature on COVID-19 vaccine acceptance among South East Asian Countries. The papers aim to assess vaccine acceptance and identify the determinants of vaccine acceptance specifically in South East Asian countries; it will add to the literature on vaccine hesitancy and help identify areas where strategies to improve vaccine acceptance are most needed. Overall, the review/meta-analysis seems very thorough and the addition of gray literature allows you to capture more in your review
We believe this is an important paper, but the framing of the intro and discussion, and the overall writing needs some revision.
In the beginning of the discussion, the paper is assuming that vaccine acceptance translates to vaccine uptake. Vaccine acceptance and vaccine uptake are two very different concepts. Vaccine acceptance is defined by “the degree to which individuals accept, question, or refuse vaccination”. Vaccine acceptance does not indicate whether a person has received a vaccine. Therefore, the authors should not make this assumption and should not use vaccination acceptance to discuss herd immunity (Discussion section paragraph 1). Later in the discussion you acknowledge that vaccine acceptance does not translate to vaccine uptake, but still compare vaccine acceptance to vaccination rates as if vaccine acceptance is an indicator of uptake, the authors should not use this comparison. Please revise this differently, the reviewers can see the connection the authors are trying to make (that if more people are likely to accept the vaccine, then that might lead to higher vaccination rates) but these are still very different concepts and should not be compared in the way the paper is comparing them.
We also have a few questions and suggestions to expand the paper on the following…
1. For the second aim, the authors aim to determine what factors influence COVID-19 vaccine acceptance. Did the authors have any prior hypotheses or justifications from previous literature to select the variables to look at? You could expand upon this in the introduction.
2. Expanding the introduction/background to better explain vaccine acceptance/hesitancy/refusal. Vaccine acceptance is a very complex issues and there are multi-level factors contributing factors such as access, time, trust, fear, perceived need/susceptibility, cost/benefits, and the influence of misinformation. The authors suggest referencing more vaccine hesitancy literature to explain the scope of this issue in the introduction.
3. What were the different scales/questions utilized to measure vaccine acceptance through out all the studies? We suggest providing some examples of the questionnaire items most commonly used, to give the reader an example of how these studies measure vaccine acceptance.
4. Can you expand on the inclusion and exclusion criteria for the review. Was there a specific time frame for your search/eligibility criteria? Did you include articles in different languages? The paper mentions that one of the searches was conducted in both English and Bahasa Indonesian. Authors should mention all inclusion and exclusion criteria and the parameters for the search more explicitly.
5. If articles in different languages were included, how did you ensure accuracy and consistency of data synthesis?
6. The last paragraph on page 3 that continues onto page 4 (starts with “Sampling methodology….” ) sounds like it could be added to the limitations paragraph. We suggest adding a sub header entitled “Limitation” to help organize the paper
7. The authors could also further expand on future research directions for assessing vaccine acceptance and determinants of vaccine acceptance in SEA countries in the conclusion to wrap up the paper.
There are also some areas where the writing needs attention.
For instance…
1. Lines 35-36: Do you mean rapid COVID-19 VACCINE development?
“In terms of public health, rapid COVID-19 development is a significant medical success, and it is 36 touted as a game changer in this pandemic.”
2. Lines 51-55: These few sentences are hard to follow. We suggest revising for clarity.
“However, eight studies, four from India 51 or Bangladesh, are not part of SEA. The other four studies originated from Indonesia and 52 Malaysia, hardly an adequate representative of SEA(26). Despite making up 8.58% of the 53 world population(27), studies focusing on COVID-19 vaccination acceptance and its 54 determinants in SEA are surprisingly rare, if there is one at all.”
3. The paper is a bit choppy in the discussion section, the use of sub-headers to organize it would be help the flow.
Reviewer 2 Report
I read with interest the study presented by Yanto and colleagues on the prevalence and determinants of Covid-19 vaccine acceptance in Southeast Asia.
The time and effort into this systematic review and meta-analysis are commendable, as well as the number of respondents included.
The study presents interesting results, contradicting many published peer-review studies.
Major comments:
- In general, scientific findings, found in research studies, can only be supported or refuted by other studies. Hence, it is OK for the current study to have one or disagreement between the determinants that influence Covid-19 vaccine acceptance. However, it is not clear why this study did not identify any determinant that affects vaccine uptake.
- The inclusion of grey literature in this meta-analysis is quite questionable and is not well justified. “Some of the grey literature is also of good quality” is an unbacked statement, and a single example (OCTA) is not enough to support it.
- It is not clear how many of the included studies are from grey literature, and whether they skewed the findings of this study.
Minor comments:
- Line 35: It would be of great addition to include citations that describe the efficacy of non-medical interventions that caused a reduction in the spread of Covid-19, e.g., doi: 10.3390/ijerph18020783
- Figure 1 (the PRISMA chart) is incredibly large and not clear. It might be worth reconsidering its size and only including the necessary information, and attach the remaining as a supplementary.
- The in-text citation and reference list is not according to the journal’s style.
- The writing in figure 2 is not clear.
- Some claims are not referenced “ It is well known that Vietnam is a model success story regarding COVID-19 vaccination uptake“
- “We only managed to find one meta-analysis that agrees with our findings, while other meta-analyses found significant associations between the variables studied and COVID-19 vaccine acceptance”. Please cite these studies.
- The sentence “Categorization of education to finish at least junior high schools and above and below junior high schools).” Is not clear. Please rephrase.
Reviewer 3 Report
The manuscript by Theo Audi Yanto et al. reviewed the COVID-19 vaccine acceptance in various countries in Southeast Asia. The authors searched various databases and excluded irrelevant or low-quality publications. Also, the authors had detailed description of methodology and reasonable and insightful interpenetrations of the data. This review is very interesting and would be important to inform people of COVID-19 vaccine acceptance in countries of Southeast Asia and determinants associated with it. The limitation of this manuscript is English language. I would suggest some minor changes as listed below.
1, “Southeast Asia” in the title while “South East Asia (SEA)” in the text. Please make it consistent.
2, Line 9 “non-existent” →”lacking”.
3, “its acceptance” →”the vaccine’s acceptance”
4, Please remove “its negative impact on” in Line 32.
5, Line 34-35 “wearing masks” →”mask mandate”.
6, Line 36 “COVID-19 development is” → “COVID-19 Vaccine Campaigns are”.
7, Line 41 “threatens”→”impedes”
8,”One author” →”It has been”
9, “higher in SEA”: higher than who?
10, “there are some strengths to our study” → “there are some strengths of our study”
11, Font size in figure 1 is not uniform.
Round 2
Reviewer 2 Report
Many thanks for addressing most of the concerns.
My last comment is that the added references are not included correctly and uniformly in the references list. Please include full citations as clearly instructed on the journal's page (https://www.mdpi.com/journal/tropicalmed/instructions#preparation):
Author 1, A.B.; Author 2, C.D. Title of the article. Abbreviated Journal Name Year, Volume, page range.
Author Response
Dear reviewer #2:
Thank you for pointing this out to us. We use EndNote for citations, and as instructed by the journal's requirement, we have downloaded the extension for ACS style and applied it to our manuscript. While the style is not exactly as outlined, we have used the journal's extension for EndNote, which should be similar to what the journal requires.
"The reference list should include the full title, as recommended by the ACS style guide. Style files for Endnote and Zotero are available."
We have no means of editing manually for the abbreviated journal name, as Endnote is an automated referencing tool. Regardless, the extension provided by the journal gave us the results manifested in our references list. We hope for the reviewer's consideration as we believe the citations included in our manuscript will also be of use (in terms of increasing the citation count).
Thank you, and we seek for kind approval.
